# Sound exposure dynamically induces dopamine synthesis in cholinergic LOC efferents for feedback to auditory nerve fibers

Jingjing Sherry Wu[1,2,3†], Eunyoung Yi[4], Marco Manca[2,3,5], Hamad Javaid[2,3,5], Amanda M Lauer[2,3,5], Elisabeth Glowatzki[1,2,3,5]*

[1]Department of Neuroscience, The Johns Hopkins University School of Medicine, Baltimore, United States; [2]The Center for Sensory Biology, The Johns Hopkins University School of Medicine, Baltimore, United States; [3]The Center for Hearing and Balance, The Johns Hopkins University School of Medicine, Baltimore, United States; [4]College of Pharmacy and Natural Medicine Research Institute, Mokpo National University, Muan-gun, Republic of Korea; [5]Department of Otolaryngology-Head and Neck Surgery, The Johns Hopkins University School of Medicine, Baltimore, United States

**Abstract** Lateral olivocochlear (LOC) efferent neurons modulate auditory nerve fiber (ANF) activity using a large repertoire of neurotransmitters, including dopamine (DA) and acetylcholine (ACh). Little is known about how individual neurotransmitter systems are differentially utilized in response to the ever-changing acoustic environment. Here we present quantitative evidence in rodents that the dopaminergic LOC input to ANFs is dynamically regulated according to the animal's recent acoustic experience. Sound exposure upregulates tyrosine hydroxylase, an enzyme responsible for dopamine synthesis, in cholinergic LOC intrinsic neurons, suggesting that individual LOC neurons might at times co-release ACh and DA. We further demonstrate that dopamine down-regulates ANF firing rates by reducing both the hair cell release rate and the size of synaptic events. Collectively, our results suggest that LOC intrinsic neurons can undergo on-demand neurotransmitter re-specification to re-calibrate ANF activity, adjust the gain at hair cell/ANF synapses, and possibly to protect these synapses from noise damage.

*For correspondence:
eglowat1@jhmi.edu

Present address: †Department of Neurobiology, Harvard Medical School, Boston, United States

Competing interests: The authors declare that no competing interests exist.

## Introduction

In our daily lives, we are routinely exposed to a constantly changing acoustic environment, for example while walking from the office to the cafeteria or from home to our commute. Changes in the acoustic scene require rapid analysis of relevant acoustic information and adjustment of sound input gain. The auditory efferent system is critical for top-down modulation of sensory flow. Sound information is collected by cochlear hair cells in the inner ear and transmitted to the brain via auditory nerve fibers (ANFs). The olivocochlear (OC) system is the final and mandatory step for direct modulation of hair cell and ANF activity in the peripheral hearing organ, the cochlea.

In rodents, one subset of OC neurons, the lateral olivocochlear (LOC) neurons, originate in and around the lateral superior olive (LSO) in the brainstem and send axons that synapse onto the unmyelinated dendrites of ANFs underneath the inner hair cells (IHCs) (*Figure 1A*; *Warr and Guinan, 1979*). These synapses are strategically located before the spike initiation zone on the myelinated peripheral ANF axons (*Hossain et al., 2005*), allowing them to directly modulate the hair cell

**eLife digest** Every day, we hear sounds that might be alarming, distracting, intriguing or calming – or simply just too loud. Our hearing system responds to these acoustic changes by fine-tuning sounds before they enter the brain. For example, if a noise is too loud, the volume can be turned down by dampening the signals nerve fibers in the ear send to the brain. This is thought to reduce the damage loud sounds can cause to the sensory organ inside the ear.

A set of nerve cells located at the base of the brain called the lateral olivocochlear (LOC) neurons coordinate this adjustment to different volumes and sounds. When these neurons receive information on external sounds, they signal back to the hearing organs and adjust the activity of auditory nerve fibers that communicate this information to the brain. LOC neurons use a diverse range of molecules to modify the activity of auditory nerve fibers, including the 'feel-good' neurotransmitter dopamine. But it is unclear what role dopamine plays in this auditory feedback loop.

To find out, Wu et al. studied the hearing system of mice that had been exposed to different levels of sound. This involved imaging LOC neurons stained with a marker for dopamine and measuring the activity of nerve fibers in the inner ear. The experiments showed that LOC neurons in mice that had recently been exposed to sound were covered in an enzyme that is essential for making dopamine. The louder the sound, the more of this enzyme was present, suggesting that the amount of dopamine released depends on the volume of the sound.

LOC neurons release another neurotransmitter called acetylcholine, which stimulates activity in auditory nerve fibers. Wu et al. found that dopamine and acetylcholine are released from the same group of LOC neurons. However, dopamine had the opposite effect to acetylcholine and reduced nerve activity. These findings suggest that by controlling the mixture of neurotransmitters released, LOC neurons are able to fine-tune the activity of auditory nerve fibers in response to acoustic changes.

This work provides a new insight into how our hearing system is able to perceive and relay changes in the sound environment. A better understanding of this auditory feedback loop could influence the design of implant devices for people with impaired hearing.

transmitted postsynaptic activity in the ANFs and thereby affect firing rates and neural coding in the auditory nerve.

LOC neurons utilize a diverse cohort of neurotransmitters and neuromodulators, including γ-aminobutyric acid (GABA), calcitonin gene-related peptide (CGRP), opioid peptides, acetylcholine (ACh) and dopamine (DA) (*Ciuman, 2010*; *Darrow et al., 2006b*; *Eybalin, 1993*; *Reijntjes and Pyott, 2016*; *Sewell, 2011*; *Vetter et al., 1991*). However, even for the better investigated cholinergic and dopaminergic LOC pathways, there is only limited and sometimes contradictory knowledge available regarding their function and little is known about the underlying mechanisms for modulating afferent activity (*Arnold et al., 1998*; *d'Aldin et al., 1995*; *Felix and Ehrenberger, 1992*; *Garrett et al., 2011*; *Maison et al., 2012*; *Maison et al., 2010*; *Niu and Canlon, 2006*; *Nouvian et al., 2015*; *Oestreicher et al., 1997*; *Ruel et al., 2001*; *Sun and Salvi, 2001*).

LOC neurons have been divided into two subgroups, based on morphological criteria (*Figure 1A and B*; *Brown, 1987*; *Vetter and Mugnaini, 1992*; *Warr et al., 1997*). In mice, the somata of LOC 'shell' neurons are located in the periolivary regions around the LSO. Their axons usually bifurcate upon entering the organ of Corti and travel extensively along the cochlear spiral, forming sparse terminals along the way. The somata of LOC 'intrinsic' neurons reside within the LSO. When reaching the cochlea, their axons usually turn in one direction, and form a patch with a high density of bouton terminals along the cochlear coil. The majority of LOC intrinsic neurons are cholinergic (*Maison et al., 2003*; *Safieddine and Eybalin, 1992*; *Figure 1B*). In mice, it is believed that dopaminergic LOC neurons form a separate neurochemical group and are mainly shell neurons (*Darrow et al., 2006b*; *Figure 1B*). However, in guinea pig, dopaminergic neurons overlap with cholinergic LOC intrinsic neurons (*Safieddine et al., 1997*). Several studies have perfused transmitters into the cochlea and recorded ANF activity *in vivo*. These studies suggest that ACh can increase and dopamine can decrease ANF firing rates (*Arnold et al., 1998*; *d'Aldin et al., 1995*; *Felix and*

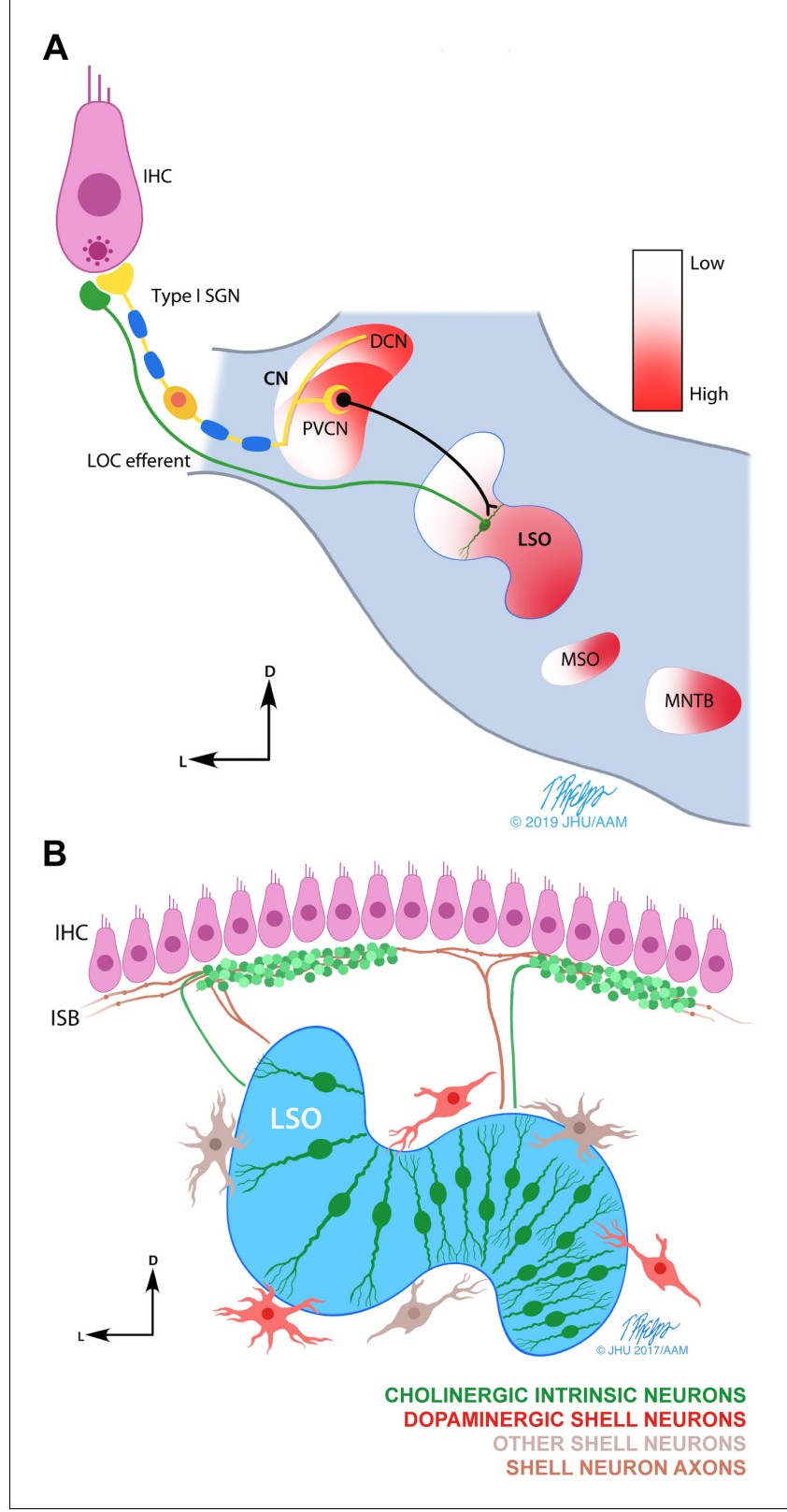

**Figure 1.** Schematic drawings of the LOC efferent system. (**A**) Line drawing illustrating the postulated LOC acoustic reflex (*Guinan, 2011*). The inner hair cell (IHC) transmits the sound signal via auditory nerve fibers (ANFs) and via the posteroventral subdivision of the cochlear nucleus (PVCN) to the lateral olivocochlear (LOC) neurons in and around the lateral superior olive (LSO) (*Thompson and Thompson, 1991*). LOC efferents signal back to the

*Figure 1 continued on next page*

*Figure 1 continued*

cochlea to the ANFs endings directly under the IHCs. The tonotopic frequency map (from low to high) found in the cochlea, is also present in multiple auditory brainstem nuclei (shown in color gradient). Other nuclei within the superior olivary complex: medial superior olive (MSO) and medial nucleus of the trapezoid body (MNTB). (**B**) Illustration summarizing the current understanding of the anatomy and neurochemistry of LOC intrinsic and shell neurons in mice (modified from *Warr et al., 1997*). LOC intrinsic neurons reside within the LSO, whereas LOC shell neurons are located on the outskirts of the LSO. Intrinsic neuron axons cover a short distance along the inner spiral bundle (ISB) and form dense bouton terminals. Shell neuron axons usually bifurcate and cover an extensive distance along the cochlear spiral with *en passant* swellings. In mice, LOC intrinsic neurons have been shown to be cholinergic (*Maison et al., 2003*), while a separate group of LOC shell neurons has been shown to be dopaminergic (*Darrow et al., 2006b*). The higher density of intrinsic neurons in the high frequency region of the LSO is based on studies in guinea pig and rat (*Kaiser et al., 2011*; *Radtke-Schuller et al., 2015*; *Warr et al., 1997*).

© 2019 Tim Phelps, JHU AAM. Illustrations in panels A and B: Tim Phelps © 2019 JHU AAM (Department of Art as Applied to Medicine, Johns Hopkins University School of Medicine), published with permission. These illustrations are not covered by the CC-BY 4.0 licence and may not be separated from the article.

---

*Ehrenberger, 1992*; *Nouvian et al., 2015*; *Oestreicher et al., 1997*; *Ruel et al., 2001*; *Ruel et al., 2006*). It has been proposed that individual neurotransmitters differentially modulate the 'set point' of ANFs in one or the other direction and contribute to generating a continuum of spontaneous activities (*Ciuman, 2010*; *Le Prell et al., 2003*; *Nouvian et al., 2015*).

Several pieces of indirect evidence suggest that LOC neurons respond to sound (*Adams, 1995*; *Drescher et al., 1983*; *Eybalin et al., 1987*; *Thompson and Thompson, 1991*). It has been proposed that they form a three-neuron feedback loop (*Figure 1A*), namely the 'LOC acoustic reflex' (*Guinan, 2011*). In particular, dopaminergic LOC neurons have been shown to change their innervation density after sound conditioning in guinea pig, which is thought to provide protection against ANF damage induced by noise exposure (*Niu and Canlon, 2002*). Additionally, mice deficient for specific dopamine receptors were found to be more vulnerable to noise exposure compared to wild-type mice, again suggesting a protective effect (*Maison et al., 2012*). Thus, the dopaminergic LOC efferents, which are the focus of the study here, provide an interesting candidate for investigating sound environment-dependent modulation of peripheral inputs to the auditory pathway. Such regulatory feedback may be important for better detection of signals in background noise, and additionally may protect synapses from excitotoxicity as a result of excessive noise exposure.

Here we present compelling quantitative evidence that tyrosine hydroxylase (TH), an essential enzyme for the synthesis of dopamine, hence a marker for dopaminergic fibers, is dynamically regulated according to the animal's recent history of sound exposure. The assumption is that upregulation of TH results in increased dopamine synthesis and release. In response to sound exposure, TH is upregulated specifically in central and peripheral components of <u>cholinergic LOC intrinsic neurons</u> in a frequency and sound level-dependent manner. These results imply that the same LOC neurons at times may co-release ACh and DA onto ANFs, most likely resulting in complex changes at multiple sites of the IHC afferent synapses that influence firing rates and dynamic range of ANFs. Electrophysiological data show that DA reduces ANF firing rate by two mechanisms: 1) by reducing the presynaptic release rate, and 2) by reducing the EPSC amplitude and area, thereby most likely reducing the percentage of EPSPs that activate APs. Taken together, these data suggest that LOC neurons can dynamically adjust their dopamine release to change the gain at hair cell/ANF synapses.

## Results

### TH[+] LOC efferent bouton endings appear in patches at seemingly random locations along the cochlear frequency axis

To investigate the extent of dopaminergic efferent inputs to the cochlea, dopaminergic LOC fibers were labeled in whole-mount preparations of C57BL/6J mouse cochleas by immunostaining against tyrosine hydroxylase (TH), an enzyme essential for the synthesis of dopamine. As described in previous publications, three types of TH[+] neurons were found in the cochlea: sympathetic fibers

(*Hozawa et al., 1989*; *Spoendlin and Tachtensteiger, 1967*; *Terayama et al., 1966*), apical type II afferent neurons (*Vyas et al., 2017*) and a subset of LOC efferent fibers in the inner spiral bundle (ISB) below the IHCs (*Darrow et al., 2006b*; *Figures 2* and *3*). Curiously, consistent with previous descriptions in CBA/CaJ mice (*Darrow et al., 2006b*), TH$^+$ LOC efferent bouton terminals were not homogeneously distributed in the ISB along the cochlear spiral, but appeared in distinct patches (*Figure 2A,B*), here called 'terminal regions'. Besides TH$^+$ fibers with bouton terminals, TH$^+$ fiber bundles with no obvious terminal varicosities, except for occasional *en passant* swellings, were present throughout the cochlear spiral. These fiber bundles were best identified in-between the terminal regions, here called 'spiral regions' (*Figure 2C*). This pattern of alternating terminal and spiral regions is established during postnatal development, between postnatal weeks 1 and 3 (*Figure 2—figure supplement 1*).

The tonotopic frequency map along the mouse cochlear spiral ranges from ~3 kHz in the apex to ~75 kHz in the base (*Müller et al., 2005*). To investigate whether the distribution of TH$^+$ terminal regions follows any systematic pattern, TH$^+$ terminal and spiral regions were mapped along the cochlear coil in 1–3 month-old mice (n = 6 cochleas, six mice) and compared using 'line plots' (*Figure 2D*). In these line plots, upper lines represent terminal regions and lower lines spiral regions. The apical cochlear tip was set at 0%, and the basal tip at 100% of cochlear length. Below the linear axis representing the cochlear length, as reference, a logarithmic map of ANF characteristic frequency is plotted, based on *Müller et al. (2005)*. As reflected in the average line plot of all six cochleas (*Figure 2D*, bottom), terminal regions covered the base of the cochlea (80–100% of cochlear length) with a higher probability than the 0–80% of cochlear length. Otherwise, terminal patches seemed to appear randomly, with no systematic pattern regarding location or length of individual patches.

## TH$^+$ terminal regions are formed by a subset of cholinergic LOC intrinsic neurons that also express TH

Cholinergic and dopaminergic LOC fibers with bouton endings have different cochlear innervation patterns: dopaminergic bouton terminals appear in patches, whereas cholinergic bouton endings cover the entire cochlear spiral (*Maison et al., 2003*), suggesting that dopaminergic and cholinergic fibers either constitute two separate systems, or that dopaminergic fibers with bouton terminals represent a subset of cholinergic fibers. The appearance of individual TH$^+$ terminal regions is reminiscent of the cochlear innervation by individual cholinergic LOC intrinsic neurons (*Warr and Boche, 2003*; *Figure 1B*). Therefore, experiments were performed to test if cholinergic and dopaminergic LOC fibers overlap. TH immunostaining was performed on cochlear tissue with genetically labeled cholinergic LOC efferents. Choline acetyltransferase (ChAT) is one of the enzymes necessary for the synthesis of acetylcholine. In the knock-in *Chat$^{iresCre}$* mouse crossed to the Cre-dependent reporter line Ai3, the fluorescent marker EYFP is expressed in cholinergic neurons, including the cholinergic LOC intrinsic neurons, as confirmed by co-immunolabeling with an antibody against ChAT in the LSO (*Figure 3—figure supplement 1A–C*). TH immunostaining in *Chat$^{iresCre}$*; Ai3 cochleas showed the typical dopaminergic LOC innervation pattern, with distinct terminal regions. In this preparation, TH$^+$ terminals clearly constitute a subset of the cholinergic terminals (n = 6 cochleas, five mice) (*Figure 3A*; two brackets point to TH$^+$ terminal regions amongst cholinergic terminals). Co-immunostaining of TH with the vesicular acetylcholine transporter (VAChT) (n = 6 cochleas, six mice) (*Figure 3B*) further confirmed the cholinergic identity of TH$^+$ bouton terminals.

To determine if the morphology of individual TH$^+$ LOC neurons is consistent with that of cholinergic LOC intrinsic neurons, a knock-in *Th$^{2A-CreER}$* mouse (*Abraira et al., 2017*) crossed with the Cre-dependent reporter line Ai9 was used to sparsely label TH$^+$ neurons by the administration of a low dose of tamoxifen (*Feil et al., 2009*). The morphology of individual TH$^+$ fibers compares well with the known morphology of cholinergic intrinsic neurons: their axons travel some distance in one direction along the inner spiral bundle under the IHCs before forming a patch of dense bouton endings (*Brown, 1987*; *Vetter and Mugnaini, 1992*; *Warr et al., 1997*) (n = 8 fibers in seven cochleas, six mice) (*Figure 3C–D*). Consistently, in brainstem sections of *Chat$^{iresCre}$*; Ai3 mice, TH labeled a subset of cholinergic LOC intrinsic neurons (*Figure 3—figure supplement 1D* and Figure 5C).

Besides TH$^+$/ChAT$^+$ LOC intrinsic neurons, TH$^+$/ChAT$^-$ LOC shell neurons were also observed around the LSO (*Darrow et al., 2006b*; *Figure 3—figure supplement 1D*), and LOC fibers with a shell neuron morphology (bifurcating axons with diffuse and sparse terminals) were found in sparsely

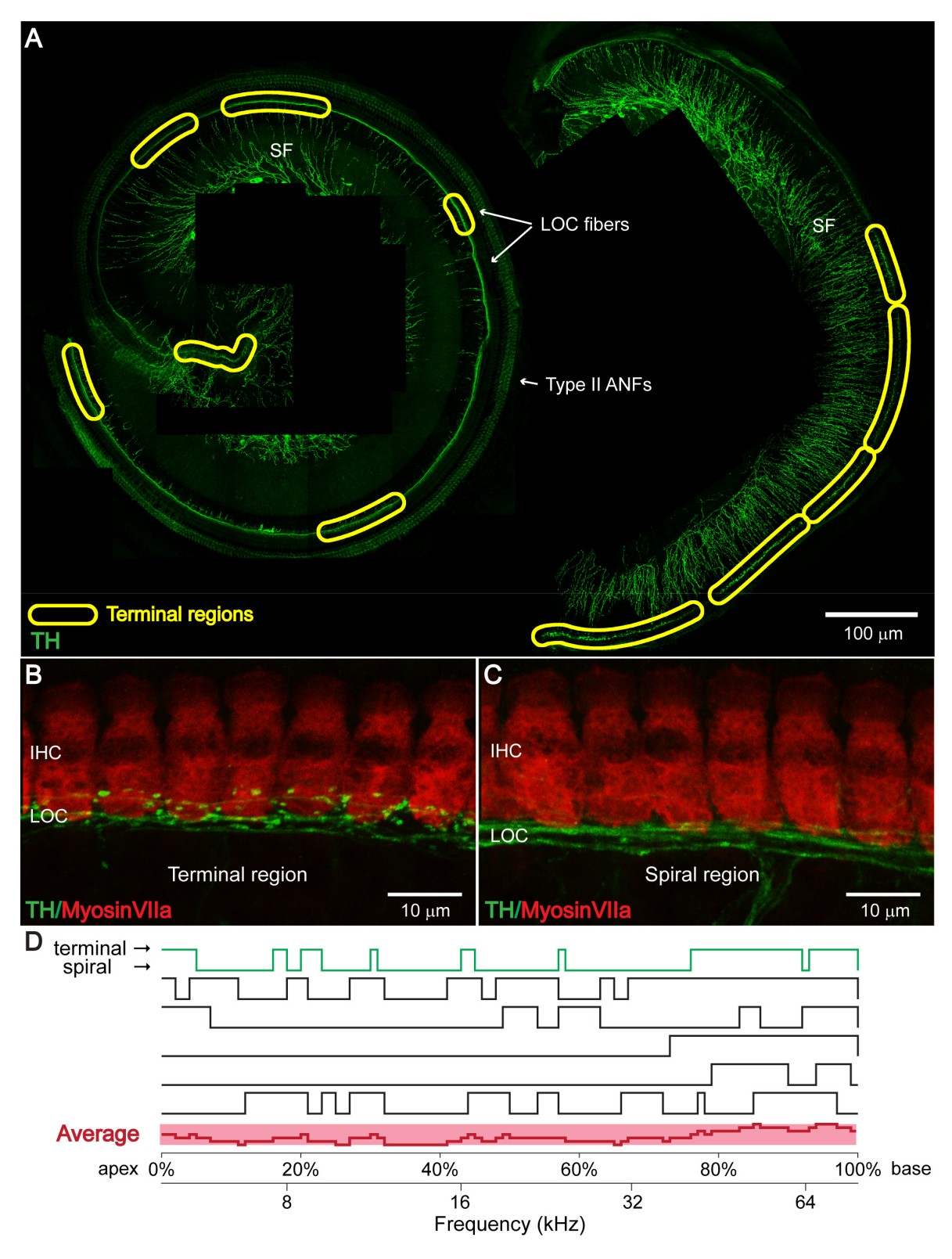

**Figure 2.** TH+ LOC efferent bouton endings appear in patches at seemingly random locations along the cochlear frequency axis. (**A**) TH immunostaining in a one-month-old cochlear whole mount preparation (left: apical half; right: basal half). Bundles of TH+ LOC efferent fibers (LOC fibers) run underneath the IHCs along the whole cochlear spiral, either with only a few swellings in 'spiral regions', or in patches with many bouton endings in 'terminal regions', as marked by yellow circles. TH also labels type II auditory nerve fibers (Type II ANFs) and sympathetic fibers (SFs). (**B** and

*Figure 2 continued on next page*

Figure 2 continued

C) Representative higher magnification images of a 'terminal region' and of a 'spiral region' in a 3-week-old cochlea. IHCs and dopaminergic LOC fibers are immunolabeled with Myosin VIIa and TH antibodies respectively. (D) Line plots along cochlear coil for six 1–3 months old cochleas indicating TH$^+$ efferent terminal regions (upper line) and spiral regions (lower line). Line plot for the representative cochlea shown in (A) is colored green. The average line plot for the six cochleas is shown in red at the bottom. The upper x-axis represents the linear distance along the cochlear spiral (with 0% at the apex and 100% at the base). The lower x-axis relates cochlear spiral location to ANF characteristic frequency. See also *Figure 2—figure supplement 1*.

The online version of this article includes the following figure supplement(s) for figure 2:

**Figure supplement 1.** Developmental Changes in TH$^+$ LOC Fiber Innervation Pattern.

labeled *Th$^{2A-CreER}$*; Ai9 mice in the cochlea (*Figure 3—figure supplement 2A*). Additionally, TH$^+$/ ChAT$^-$ fibers of an unreported morphology were observed in the cochlea (*Figure 3—figure supplement 2B*). Nonetheless, neither of these TH$^+$/ChAT$^-$fibers form terminals that cluster into discrete patches of dense bouton endings. Therefore, they were not further investigated in this study.

Together, these data suggest that TH$^+$ bouton endings in terminal regions are formed by TH-expressing cholinergic LOC intrinsic neurons.

## Sound exposure increases the percentage of cochlear spiral covered with TH$^+$ terminal regions

Results described so far (*Figures 2* and *3*) were based on cochleas harvested from mice that were raised in an institutional vivarium with a highly variable and generally noisy sound environment (*Lauer et al., 2009*; *Figure 4—figure supplement 1A*). Previous studies in guinea pig have shown that sound conditioning induces an increase in TH$^+$ fibers in the IHC region of the cochlea (*Niu and Canlon, 2002*). The highly variable TH$^+$ LOC efferent innervation patterns across the genetically homogenous WT mice therefore could result from differences in the acoustic experience of individual animals. Thus, we hypothesized that the distribution of TH$^+$ terminal patches in individual cochleas is dynamically regulated by sound.

To test this hypothesis, mice were raised in a 'low noise' vivarium with lower ambient sound levels (*Figure 4—figure supplement 1A*; *Lauer et al., 2009*), and their cochleas were immunostained for TH at the age of 8 weeks. Line plots show the coverage of TH$^+$ terminal and spiral regions in an example cochlea (*Figure 4A*, control) and for the average of 22 cochleas (11 mice) (*Figure 4B*, control). In contrast to mice raised in the institutional vivarium (*Figure 2D*), cochleas from mice raised in the 'low noise' vivarium showed only a few or no terminal regions in the apical half (no terminal patches < 30 kHz for n = 19/22 cochleas). Most of the identified terminal regions were concentrated in the most basal, high frequency region of the cochlea.

To test if sound exposure increases the coverage of TH$^+$ LOC terminal regions along the cochlear coil, mice raised in the 'low noise' vivarium were exposed to a 12 kHz-centered one-octave noise band at 110 dB SPL for 2 hr at the age of 7 weeks (see Materials and methods). The effects of this sound exposure protocol on hearing were evaluated with auditory brainstem responses (ABRs) measured on a separate set of animals. This sound exposure paradigm resulted in irreversible ABR threshold shifts to levels > 85 dB SPL on average for the tested frequency range (clicks and 8–32 kHz tones) (*Figure 4—figure supplement 1B*). 7–10 days after noise exposure, cochleas were immunostained for TH (n = 20 cochleas, 11 mice). The delay of at least one week after noise exposure for immunostaining was chosen to allow adequate time after noise exposure for the synthesis, transport, and accumulation of TH proteins in the axonal terminals of LOC neurons, to a level that can be detected by immunostaining in the cochlear epithelium. Compared to control, after sound exposure, the apical half of the cochlear coil at and above the frequency range of the noise band showed a significantly increased likelihood to be covered by TH$^+$ terminal regions (*Figure 4A,B*, exposed). Interestingly, there was a paradoxical significant decrease in the number of terminal patches in the basal part of the cochlea, at frequencies > 60 kHz (*Figure 4B*, *Figure 4—source data 1*). Nevertheless, the total percentage of cochlear spiral covered by TH$^+$ terminal regions increased significantly, about 5-fold, in sound exposed mice compared to control littermates (median ± IQR, control: 0.1 ± 0.06, exposed: 0.47 ± 0.27) (*Figure 4C*).

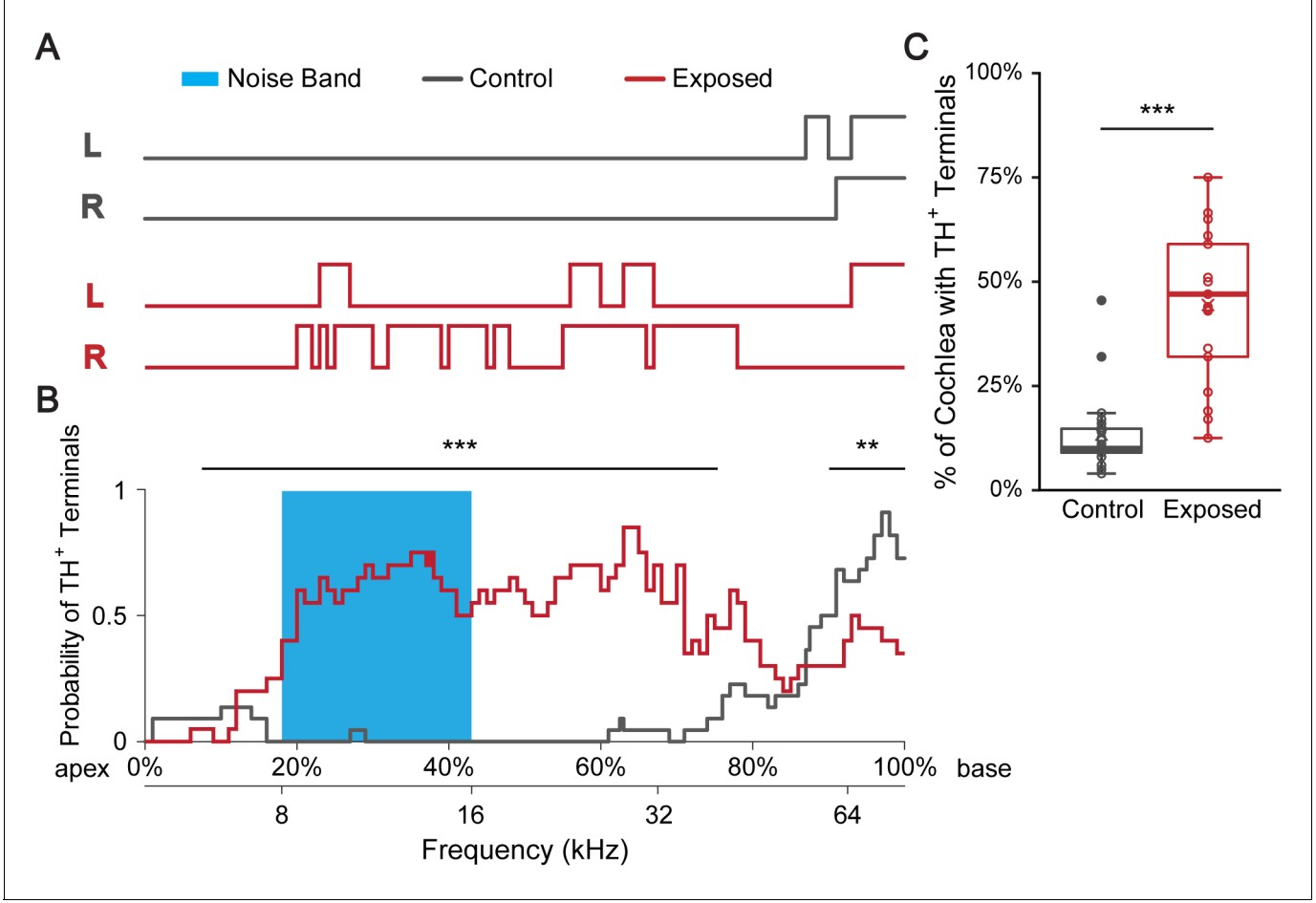

**Figure 4.** Sound exposure increases the percentage of cochlear spiral covered with TH+ terminal regions. (A) Line plots indicate TH+ terminal regions (upper lines) versus TH+ spiral regions in a representative control mouse raised in a 'low noise' vivarium (gray) and in a mouse that was exposed to a 12 kHz-centered one-octave noise band at 110 dB SPL for 2 hr (red). Immunolabeling for TH was performed 7–10 days after sound exposure. Left (L) and right (R) cochleas are shown. The upper x-axis (shown in B) represents the linear distance along the cochlear spiral (with 0% at the apex and 100% at the base). The lower x-axis relates cochlear spiral location to ANF characteristic frequency. (B) A plot showing the averaged probability for a cochlear region to be covered by TH+ terminals at the cochlear frequencies plotted on the x-axis for control (gray, n = 22 cochleas, 11 mice) and sound exposed mice (red, n = 20 cochleas, 11 mice). Blue region marks the frequency range of the noise band. For the 7.5–75.5% of the cochlear length from the apex (~6–40 kHz), sound exposed mice are significantly more likely to have TH+ bouton endings compared to control. For the 90–100% of the cochlear length from the apex (~60–75 kHz), sound exposed mice are significantly less likely to have TH+ bouton endings compared to control. Repeated measures binary logistic regression using generalized estimating equations (GEE); ***p<0.001, **p<0.01. Data for individual cochleas are provided in *Figure 4—source data 1*. (C) Sound exposed mice (n = 21 cochleas, 11 mice) showed on average 31% more cochlear coverage by TH+ terminal regions compared to their littermate control mice (n = 22 cochleas, 11 mice). Linear mixed model with sound exposure as the fixed effect and repeated measures from two ears of the same mouse; ***p<0.0005. See also *Figure 4—figure supplement 1*.

The online version of this article includes the following source data and figure supplement(s) for figure 4:

**Source data 1.** Regions of Cochlear Spiral covered by TH+ LOC terminal regions in Control and Sound Exposed (2hr 110 dB SPL) mice.

**Figure supplement 1.** Sound Levels Measured in the Institutional and a 'Low Noise' Vivarium and ABR Threshold and Threshold Shifts Induced by Sound Exposure Protocols.

## Sound exposure increases the number of TH+ cholinergic intrinsic neurons in the LSO

Though multiple potential mechanisms could account for increased cochlear coverage of TH+ terminal regions after sound exposure, the most straightforward explanation is an increase in the fraction

of cholinergic LOC intrinsic neurons that express TH. To test this hypothesis, parallel to analyzing the effects of sound exposure in the periphery (*Figure 4*), the number of TH$^+$ LOC neurons in the LSO was quantified in brainstem sections from the same set of mice. Compared to control littermates, the number of TH$^+$ LOC intrinsic neurons in sound exposed mice increased significantly, about 5-fold (median ± IQR per LSO, control: 11 ± 7, exposed: 47 ± 35) (*Figure 5A,B and D*). This 5-fold change is remarkably comparable to the increase in the percentage of cochlear spiral covered by TH$^+$ terminal regions, suggesting that the LOC neurons that became TH$^+$ due to sound exposure, covered additional length along the ISB in the cochlea.

Secondly, to verify that after sound exposure TH was in fact upregulated in cholinergic LOC intrinsic neurons, a subset of sound exposure experiments was performed on mice with genetically labeled cholinergic neurons (*Chat$^{iresCre}$*; Ai9 mice). Sound exposure did not impact the number of cholinergic LOC intrinsic neurons. The median number of genetically labeled ChAT$^+$ LOC neurons was not significantly different after sound exposure (422, n = 6 cochleas, 3 mice) compared to control littermates (433, n = 8 cochleas, 4 mice) (Mann-Whitney U test, p=0.491). After sound exposure, immunolabeled TH$^+$ neurons represented a subset (16 ± 4%) of the cholinergic LOC neurons. Again, similar to the whole dataset (*Figure 5A,B,D*), the percentage of TH$^+$ cholinergic LOC neurons in sound exposed mice was ~5 times higher compared to control (16 ± 4% versus 3 ± 1%) (Welch t-test, p<0.0005) (n = 6 LSOs, 3 mice; *Figure 5C*). In comparison, after sound exposure, no obvious change was observed in the number of TH$^+$/ChAT$^-$ LOC shell neurons (not quantified).

To test if TH expression affects the cholinergic identity of LOC terminals, after sound exposure, we immunostained for ChAT, in a *Chat$^{iresCre}$*; Ai9 mouse. Most TH$^+$ bouton terminals were positive for ChAT, both by genetic labeling (which indicates their original identity) and immunolabeling (which represents expression at the time of tissue harvesting), suggesting that when TH expression is upregulated, the cholinergic phenotype of existing ChAT$^+$ LOC neurons was maintained, for at least 7–10 days after sound exposure (*Figure 5—figure supplement 1A–B*).

In summary, sound exposure induces the expression of TH in previously TH$^-$ cholinergic LOC intrinsic neurons. The schematic of cochlear innervation by subtypes of LOC fibers has been modified to reflect these results (*Figure 5E*).

## The number of TH$^+$ LOC intrinsic neurons in the LSO is dynamically regulated by sound

To test whether TH expression in cholinergic LOC intrinsic neurons can be dynamically regulated by sound, a less damaging, and partially reversible sound exposure protocol was applied. 7-week-old mice were exposed to a 12 kHz-centered one-octave noise band at 90 dB SPL for 12 hr each day for 5 consecutive days. One week after starting sound exposure, ABRs showed significant threshold shifts to ~53 dB SPL for clicks and 55–90 dB SPL for 12–32 kHz tones (no significant change at 8 kHz) (*Figure 4—figure supplement 1C*). Three weeks after starting sound exposure, the ABR threshold shift had reversed completely for clicks and 12–16 kHz tones, and partially for 24 and 32 kHz tones (*Figure 4—figure supplement 1C*).

At one week after the sound exposure protocol had been initiated, the number of TH$^+$ LOC intrinsic neurons increased significantly in sound exposed mice compared to control littermates by about 2-fold (median ±IQR, control: 15 ± 11, exposed: 31 ± 46) (*Figure 5D*). This is about half of the increase that was found earlier with the more damaging sound exposure paradigm. Interestingly, two weeks later (three weeks after initiating the sound exposure protocol), the number of TH$^+$ LOC intrinsic neurons in sound exposed mice had returned to levels that were not significantly different from those of control littermates (median ±IQR, control: 15 ± 11, exposed: 19 ± 9) (*Figure 5D*). These results suggest that TH expression is dynamically regulated according to the animal's acoustic experience.

To demonstrate dynamic expression of TH within individual LOC intrinsic neurons over time, mouse genetic tools were used in combination with the variable acoustic environment in the institutional vivarium. Tamoxifen administration induces reporter protein (EYFP) expression in TH$^+$ LOC neurons in *Th$^{2A-CreER}$*; Ai3 mice, providing permanent labeling of LOC neurons that express TH around the time of tamoxifen administration. On the other hand, TH immunostaining, performed 1–3 weeks after tamoxifen injection will label TH-expressing LOC neurons at the time of tissue harvesting. Because of the highly variable acoustic environment in the institutional vivarium, it is expected that different sets of LOC intrinsic neurons express TH at different times. Indeed, we observed some

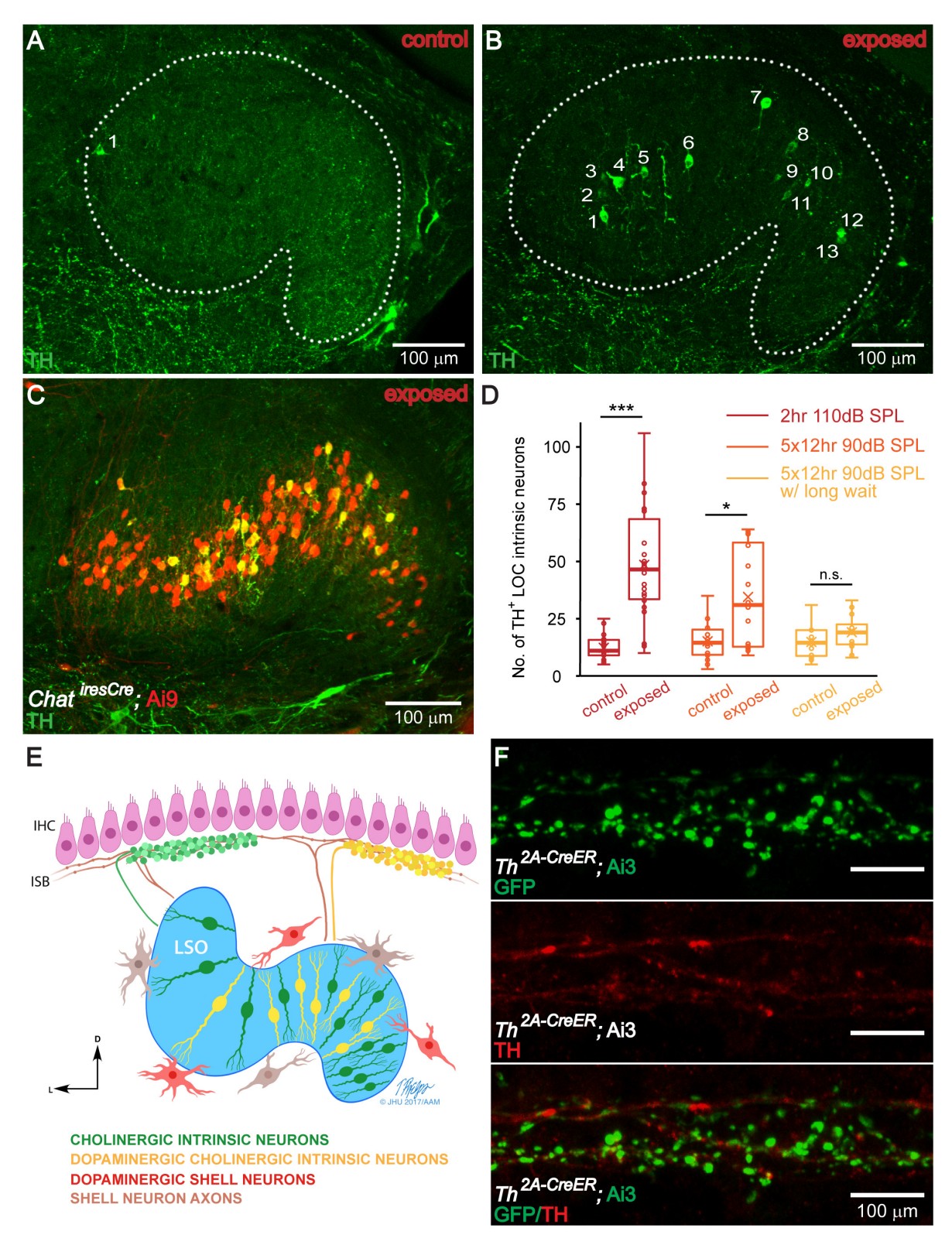

**Figure 5.** The number of TH+ LOC intrinsic neurons in the LSO is dynamically regulated by sound. (**A and B**) Representative images of TH immunolabeled 50 µm brain slices of the LSO region from a control (**A**) and a sound exposed mouse (B, 2 hr exposure). Sound exposure occurred at 7 weeks of age, and immunolabeling at 8 weeks. Each number identifies a TH+ LOC intrinsic neuron. (**C**) A representative brain slice of the LSO region from a sound exposed mouse (2 hr exposure) expressing tdTomato (Ai9) driven by *Chat^iresCre^* to label cholinergic LOC intrinsic neurons (red). Co-
*Figure 5 continued on next page*

*Figure 5 continued*

labeling with TH immunostaining (green) demonstrates that TH expression is found in existing cholinergic LOC intrinsic neurons. (D) Box plots demonstrating the number of TH$^+$ LOC intrinsic neurons identified within LSO from either side of the brain, in control versus sound exposed mice for three sets of experiments. For both, 2 hr 110 dB SPL exposure (red) (control: n = 22 LSOs, 11 mice, exposed: n = 22 LSOs, 11 mice) and 5 × 12 hr 90 dB SPL exposure (orange) (control: n = 14 LSOs, 8 mice, exposed: n = 16 LSOs, 8 mice), the exposed groups have significantly larger numbers of TH$^+$ LOC neurons than control groups, when examined one week from the beginning of the exposure (2 hr 110 dB SPL exposure, control: 11 ± 7, exposed: 47 ± 35) (5 × 12 hr exposure, control: 15 ± 11, exposed: 31 ± 46). In contrast, when examined three weeks from the beginning of the 5 × 12 hr 90 dB SPL exposure (yellow) (control: n = 12 LSOs, 6 mice, exposed: n = 12 LSOs, 6 mice), the number of TH$^+$ LOC neurons is not significantly different between exposed and control groups (5 × 12 hr exposure w/long wait, control: 15 ± 11, exposed: 19 ± 9). Linear mixed model with sound exposure as the fixed effect and a random intercept for each mouse to account for the correlation among two measurements from the same mouse, ***p<0.0005; *p<0.05; n.s. not significant p>0.05. Presented data indicate median ±IQR per LSO. (E). Modified schematic drawing demonstrating the hypothesis that after sound exposure, a subset of cholinergic LOC intrinsic neurons will become dopaminergic and cholinergic. (F). An apical cochlear segment of a Th$^{2A-CreER}$; Ai3 mouse raised in the institutional vivarium. Tamoxifen was administrated between 1–3 weeks. The mouse was sacrificed at the age of 1 month. Comparing mouse line labeling (top, a GFP antibody recognizes the EYFP reporter protein) and TH immunostaining (middle) suggests that some of the LOC efferent terminals that expressed *Th* at the time of tamoxifen injection did not express *Th* anymore at 1 month. See also *Figure 5— figure supplement 1*.

The online version of this article includes the following figure supplement(s) for figure 5:

**Figure supplement 1.** Dynamic expression of TH in LOC Fiber Bouton Endings.

LOC bouton terminals that were labeled for TH genetically, but not by immunostaining (n = 4 cochleas, 3 mice) (*Figure 5F*), suggesting that these terminals expressed TH around the time of tamoxifen injection, but no longer at the time of tissue harvesting. However, genetic labeling and TH immunolabeling often overlapped at the base, even when the times of tamoxifen injection and tissue harvesting were several weeks apart (*Figure 5—figure supplement 1C*), suggesting that the TH$^+$ LOC intrinsic neurons that innervate the basal region of the cochlea have a more stable TH expression. These results suggest that the dynamic expression of TH can be found in a 'common' acoustic environment for laboratory mice.

## Dopamine modulates auditory nerve fiber activity at the IHC afferent synapse by two distinct mechanisms

Previously, the effects of DA on ANF activity have been investigated with *in vivo* extracellular ANF recordings from guinea pig by perfusion of artificial perilymph containing DA into the inner ear (*Oestreicher et al., 1997*; *Ruel et al., 2001*). These studies suggest that DA reduces spontaneous and sound-evoked ANF activity. To investigate how DA modulates ANF activity at the cellular level, patch clamp recordings were performed at the bouton endings of ANFs directly underneath the IHCs in acutely excised rat apical cochlear coils (*Glowatzki and Fuchs, 2002*; *Grant et al., 2010*). Such recordings monitor synaptic activity at individual glutamatergic hair cell ribbon synapses, representing all the peripheral input an ANF receives. For better success rates of these technically challenging recordings, rats were used instead of mice. Recordings were performed at an age range (15–31 postnatal days), when properties of subgroups of ANFs with low to high spontaneous rates have mostly developed (*Taberner and Liberman, 2005*; *Wu et al., 2016*).

### Dopamine reduces the firing rate in ANF endings

In a first step, to monitor spike rates, extracellular loose patch recordings were performed from afferent bouton endings (*Wu et al., 2016*). Spike rates were analyzed before, during DA application (1 mM; 3–5 min), and after wash. A separate set of control experiments was performed, where the control solution was applied instead of DA. Qualitatively similar to *in vivo* studies, spikes rates in ANFs were significantly reduced by 39% on average in DA (n = 22, *Figure 6A, C and D*), whereas no significant change occurred when switching to control solution (n = 23, *Figure 6B and D*).

## Dopamine reduces the EPSC amplitude, and thereby most likely reduces the percentage of EPSPs that activate an AP

In only three of many attempted current clamp recordings, could EPSPs and APs be distinguished based on their size and waveform and where event numbers were large enough for analysis (*Figure 6E*; AP: arrow; EPSP: arrowhead). Although too small a dataset for providing solid proof, current clamp results for three ANFs are reported here, as they are instructive for suggesting a possible mechanism. At the IHC afferent synapse, it can be assumed that individual APs are activated by individual supra-threshold EPSPs (*Rutherford et al., 2012*). In the three current clamp recordings, in DA, the rate of APs decreased (ANF #1: by 97%; ANF #2: by 70%; ANF #3: by 59%; *Figure 6—figure supplement 1A*). Interestingly, in two of these recordings, the percentage of APs relative to the number of all events (EPSPs + APs) decreased in DA (ANF #1: from 84% to 50% in DA; ANF #2: from 53% to 13% in DA) (*Figure 6E*; *Figure 6—figure supplement 1B*); in other words, in DA, EPSPs were less effective in activating APs. In these two recordings, ANF membrane potentials and AP thresholds remained unchanged during DA application (*Figure 6—figure supplement 1C,D*). The reduced efficiency for activating APs in DA could be due to a reduction in EPSP amplitude/charge. To test this hypothesis, the effect of DA on EPSC waveform was investigated in voltage clamp recordings. EPSCs showed a wide range of amplitudes ranging up to 80-fold in individual recordings, due to a specialized release mechanism that has been reported earlier at IHC ribbon synapses (*Chapochnikov et al., 2014*; *Glowatzki and Fuchs, 2002*; *Grant et al., 2010*; *Li et al., 2009*; *Figure 6F*). Analysis of the exemplary recording in *Figure 6F* is reported in *Figure 6—figure supplement 1E–G*. *Figure 6G,H*, and *Figure 6—figure supplement 1H* summarize results for all voltage clamp recordings (n = 6). In DA, EPSC amplitude and EPSC area (charge) were significantly and reversibly reduced, by ~19 ± 14% and by 19 ± 9% (mean ± SD), respectively (*Figure 6G–H*), whereas EPSC half-width remained unchanged (*Figure 6—figure supplement 1G–H*). These data suggest that it is the reduced EPSP amplitude/charge that results in less effective activation of APs in DA. The combined reduction of EPSC amplitude and area, as well as the stable half-width in DA, makes it unlikely that the underlying mechanism for reducing the EPSC amplitude is based solely on presynaptic desynchronization of release events within individual EPSCs (*Chapochnikov et al., 2014*; *Grant et al., 2010*). Further experiments are needed to separate if the effect of DA on EPSC amplitude/area occurs presynaptically, in the IHC, and/or postsynaptically, in the ANF ending.

## Dopamine reduces the rate of release at the IHC afferent synapse

Besides reducing EPSC size, DA additionally decreased the rate of release at the IHC afferent synapse. To demonstrate this effect, data from current clamp and voltage clamp experiments were pooled. For current clamp recordings, APs and EPSPs, were both included as representing release events. DA application (1–2 mM) irreversibly reduced the rate of events by about half (median ±IQR, control: 11.5 ± 13.3 spikes/s, DA: 6.4 ± 10.0 spikes/s, wash: 6.9 ± 14.4 spikes/s; n = 13) (*Figure 6E,F, I*). This effect of DA on release rate was irreversible, whereas the effects on EPSC amplitude and area were reversible, suggesting that DA operates via two different mechanisms that both contribute to reducing the ANF firing rate.

In summary, the electrophysiological data show that DA reduces ANF firing via two mechanisms, one of which acts presynaptically and affects the release rate.

## Discussion

### Sound exposure dynamically regulates TH expression in LOC intrinsic neurons

This study provides several lines of evidence that sound exposure regulates TH expression in LOC intrinsic neurons in a level-dependent and dynamic manner. Mice raised in a 'low noise' vivarium showed significantly less TH$^+$ LOC terminals in the cochlea compared to mice raised in the noisier institutional facility (*Figure 4*). Damaging sound exposure increased the number of TH$^+$ LOC neurons about 5-fold, and a less damaging sound exposure protocol caused a smaller increase (~2 fold) in TH$^+$ LOC neurons (*Figure 5D*). In addition, 3 weeks after the less damaging sound exposure, the ABR thresholds of the exposed mice partially returned to control levels, along with the number of TH$^+$ LOC neurons. In *Th$^{2a-CreER}$*; Ai3 mice, a comparison of reporter protein expression and antibody

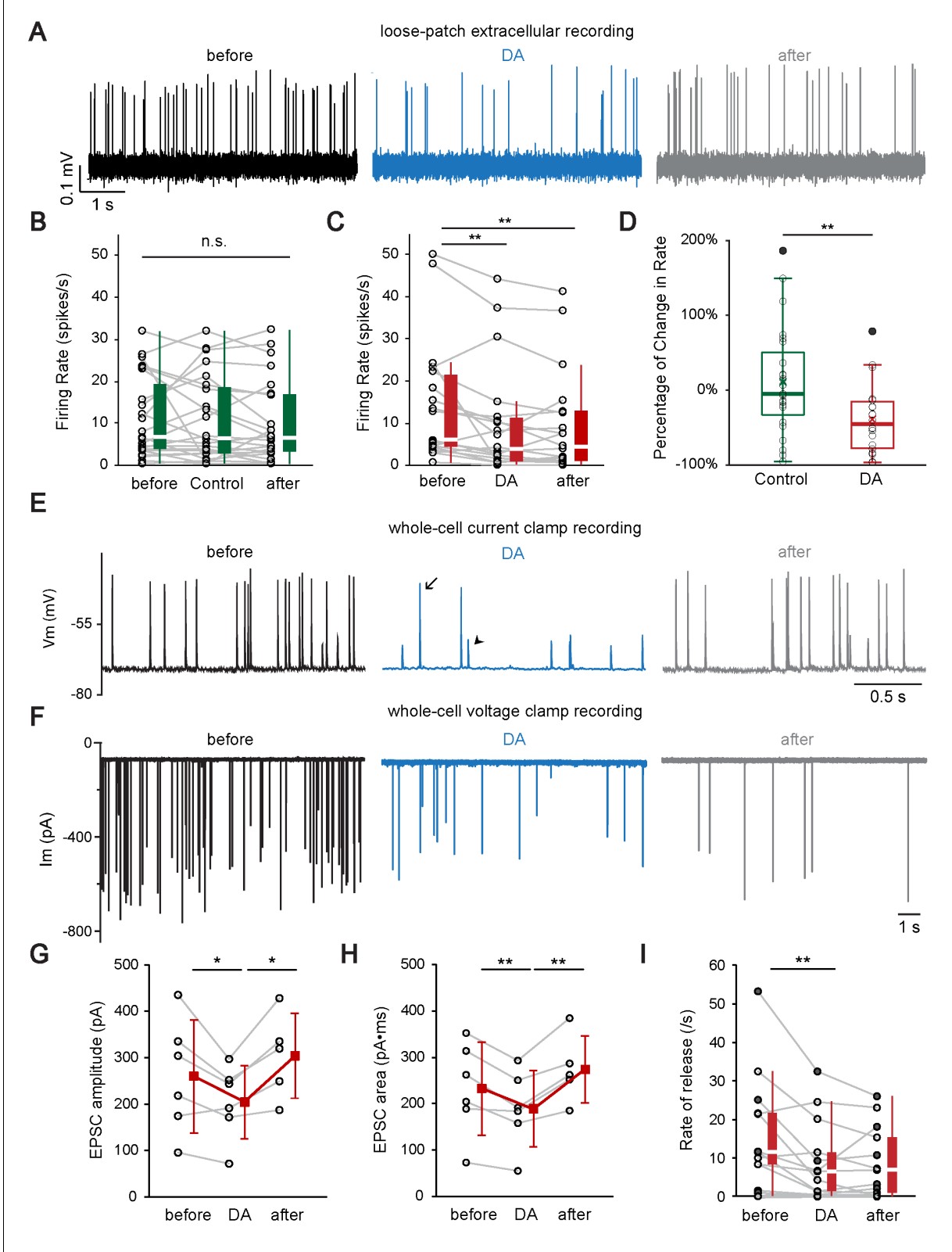

**Figure 6.** Dopamine modulates auditory nerve fiber activity at the IHC afferent synapse by two distinct mechanisms. (A) A representative loose-patch extracellular recording demonstrates that DA (dopamine, 1 mM) reduced the firing rate of the ANF. (B) Profile plots for individual recordings of control experiments (n = 23). Box plots are shown to the right of individual data. Switching from extracellular solution to Control solution, which contains 0.1% sodium ascorbate, the antioxidant used in DA solution, did not cause any significant change in firing rate. Friedman test, n.s. not significant. (C) Profile
*Figure 6 continued on next page*

Figure 6 continued

plots for individual recordings with DA application (n = 22). Box plots are shown to the right of individual data. DA significantly reduced the firing rate. Friedman test followed by Dunn's multiple comparisons test, **p<0.01. Note that one fiber with a high spontaneous rate (~90 spikes/s) was omitted from the individual profile plots for better visualization of other fibers, but is included for statistical analysis. (D) Box plots of the percentage of change in firing rate during the application of Control solution (B) or DA (C). Two-tailed independent t-test, **p<0.01. (E) A representative current clamp recording demonstrates that DA (1 mM) reduces the rate of synaptic events and decreases the percentage of EPSPs (arrowhead) that activate APs (arrow). (F) A representative voltage clamp recording demonstrates that DA (2 mM) application reduced the rate and amplitude of EPSCs. (G–H) Profile plots showing changes in the (G) EPSC amplitude and (H) EPSC area during DA application for 6 voltage clamp recordings. Mean ± SD (red squares) is shown to the right of individual data (open black circles). In order to utilize all available data points, two-tailed paired t-tests were performed separately between 'before' and 'DA' group (n = 6) and between 'DA' and 'after' group (n = 5), *p<0.05, **p<0.01. (I) Profile plot of synaptic rate for individual fibers before (before), during (DA) and after (after) dopamine application for 6 current clamp recordings (filled circles; both APs and EPSPs were counted) and 7 voltage clamp recordings (open circles). Box plots are shown to the right of individual data. One extreme outlier was excluded from this analysis. One-way repeated-measure ANOVA on rank transformed data, followed by post hoc pairwise comparison with a Bonferroni adjustment, **p<0.01. See also *Figure 6—figure supplement 1*.

The online version of this article includes the following figure supplement(s) for figure 6:

**Figure supplement 1.** Dopamine reduces the EPSC amplitude, and thereby most likely reduces the percentage of EPSPs that activate an AP.

labeling showed that individual neurons expressed different levels of TH at different points in time (*Figure 5E*).

Consistent with these results, previous studies have shown that dopamine metabolites increased in the rat cochlea after sound exposure proportionally to sound intensity (*Gil-Loyzaga et al., 1993*). In addition, an increase in TH immunoreactivity was observed in the guinea pig cochlea after non-damaging sound conditioning (*Niu et al., 2004*; *Niu and Canlon, 2002*). However, in contrast to what we observed, studies in guinea pig showed that damaging sound exposure down-regulated TH immunoreactivity in both cochlea and brainstem. This discrepancy might be due to species differences or to the use of anesthesia during the damaging sound exposure, but <u>not</u> during the conditioning sound exposure by Niu and Canlon, whereas sound exposures in the study here were performed on awake animals. Suppressive effects of anesthesia on efferent function have been demonstrated at least for MOC neurons (*Chambers et al., 2012*).

For both, mice raised in the institutional and in the 'low noise' vivarium, TH$^+$ terminal regions covered more prominently the base where high frequency sounds are represented (*Figure 2D*, *Figure 4B*, control). This is most likely due to a larger number of TH$^+$ LOC intrinsic neurons projecting to the base, rather than due to an increase in the length of the terminal bouton patches formed by individual LOC intrinsic neurons that innervate the base of the cochlea. Reconstructions of single LOC intrinsic neuron axons do not show any systematic variation in the length of the terminal arbor along the cochlear spiral (*Warr et al., 1997*; *Warr and Boche, 2003*). Similarly, dopaminergic LOC neurons in the guinea pig cochlea preferentially innervate the high frequency region (*Mulders and Robertson, 2004*). However, sound level measurements both in the 'low noise' and in the institutional vivarium did not report high sound levels in the high frequency range (*Figure 4—figure supplement 1A*). One possibility is that TH expression in LOC neurons in the high frequency range is activated by mouse ultrasonic vocalizations (*Lahvis et al., 2011*). Another possibility is that LOC neurons are denser in the medial, high-frequency processing LSO limb (*Kaiser et al., 2011*; *Radtke-Schuller et al., 2015*), which could result in a higher probability of TH$^+$ LOC efferent terminals to appear in the base. Notably, after damaging noise exposure, a paradoxical decrease of innervation by TH$^+$ terminal in the basal cochlear coil was observed (*Figure 4B*). This could be due to noise-induced damage to ANF terminals and subsequent loss of efferent innervation.

## Mechanisms underlying dopaminergic modulation of ANF activity

In the excised rat cochlea *in vitro*, dopamine reduces the firing rate of ANFs on average by ~40%. These data are qualitatively similar to previous results reporting reduced ANF firing when dopamine was perfused into the guinea pig inner ear *in vivo* (*Oestreicher et al., 1997*; *Ruel et al., 2001*). The study here describes two mechanisms by which dopamine reduces ANF firing rate: (1) by a reduction in the synaptic event rate and (2) by a reduction in EPSC amplitude and area. (1) The reduction in

event rate suggests a presynaptic mechanism; hair cell release is affected by dopamine. Such a pre-synaptic downregulation of afferent activity provides an unexpected but highly effective potential strategy for avoiding glutamate-induced excitotoxic effects on afferent endings in response to sound exposure (*Kujawa and Liberman, 2009*; *Le Prell et al., 2003*; *Nouvian et al., 2015*). IHC release could be affected if DA were to act in a paracrine fashion by diffusing and binding to hypothetical dopamine receptors on the IHCs. Direct protein-protein interaction between dopamine D1A recep-tor and components of the vesicular exocytosis machinery, including otoferlin, have been demon-strated (*Selvakumar et al., 2017*), suggesting that dopamine receptors could potentially directly influence the release process. However, dopamine receptor expression has not been found in mouse or rat IHCs (*Inoue et al., 2006*; *Maison et al., 2012*), though they have been described in fish hair cells (*Drescher et al., 2010*; *Perelmuter et al., 2019*; *Toro et al., 2015*). An alternative mechanism of affecting IHC release rate could be through a dopamine receptor-dependent release of retro-grade messengers from ANFs. (2) The reduction in EPSC amplitude and area by dopamine highly likely reduces the EPSP amplitude/area and thereby a lower percentage of EPSPs may activate APs. The effect on EPSC amplitude/area could originate pre- and/or postsynaptically, for example by reducing the quantal size or influencing glutamate receptor function.

Additionally, dopamine could affect the excitability of ANFs by modulating multiple ion chan-nels in afferent dendrites. For example, studies on guinea pigs and rats suggested that DA could affect ANF firing by decreasing sodium currents (*Oestreicher et al., 1997*; *Valdés-Baizabal et al., 2015*). Finally, besides dopamine receptors on the ANFs, the LOC efferent ter-minals may contain D2 receptors (*Inoue et al., 2006*; *Maison et al., 2012*). Dopamine acting on D2 auto-receptors at the LOC efferent terminals may alter the release of dopamine and other LOC efferent neurotransmitters, further complicating the effects of dopamine. Future studies are needed to dissect the different mechanisms responsible for reducing ANF firing rates by dopa-mine released from LOC efferent fibers.

## Orchestra of ACh and DA effects on ANFs

In CBA/CaJ mice, dopaminergic LOC neurons were mainly identified as non-cholinergic shell neu-rons, located just outside of the boundaries of the LSO, suggesting that dopaminergic and choliner-gic LOC neurons exist as separate groups (*Darrow et al., 2006b*). The study here also reports non-cholinergic dopaminergic shell neurons in C57BL/6J mice. Surprisingly, TH expression is additionally found in a small percentage of cholinergic intrinsic neurons in the LSO. Cholinergic bouton endings of LOC fibers are found all along the cochlea, however, in 'patches' these endings co-express TH. Even more TH$^+$ cholinergic patches can be found after sound exposure. The co-expression of TH and ChAT suggests that potentially <u>both</u> ACh and DA are being released from the same neurons. These results drastically change our thinking about how LOC intrinsic neurons may modulate spike rates of individual ANFs.

*In vivo* experiments in the guinea pig showed that cochlear perfusion of ACh increases and DA decreases ANF firing rate (*Arnold et al., 1998*; *d'Aldin et al., 1995*; *Felix and Ehrenberger, 1992*; *Garrett et al., 2011*; *Nouvian et al., 2015*; *Oestreicher et al., 1997*; *Ruel et al., 2001*). The study here supports that DA decreases ANF firing rates, however, there are no consistent *in vitro* data available yet for effects of ACh or ACh/DA combined on ANF activity. Assuming that the two neuro-transmitter systems act in divergent ways, it has been hypothesized that the LOC efferent system sets the sensitivity of the ANFs by manipulating a 'set point' (*Nouvian et al., 2015*; *Ruel et al., 2006*). The current study extends our understanding of how this might work. With the ability to actively adjust its relative dopaminergic versus cholinergic output, the LOC system may be able to fine-tune ANF activity in response to the sound environment. Thus, we hypothesize that cholinergic outputs dominate in a low noise environment and provide a basal level of positive modulation of ANF activity as cholinergic innervation is found all along the cochlear coil, independent of the sound environment; in animals raised in quiet as well as after sound exposure. In response to sound expo-sure, DA is released additionally from cholinergic terminals in a frequency-dependent manner, to tune down ANF activity and adjust their sensitivity, and, in the case of damaging sound exposure, to possibly prevent excitotoxic effects on ANF dendrites.

The dynamic changes occurring in the neurochemical profile of the LOC efferent system based on the sound environment could explain the puzzling contradictory results of several classic LOC lesion studies. The immediate effect of such lesions could either result in an enhancement (*Darrow et al.,*

*2007*) or in depression (*Le Prell et al., 2005*) of ensemble ANF activity, which is probably due to differences in the output profile of LOC efferents at the time of the experiments. Chronic LOC lesioning reduced the basal positive tone provided by LOC efferents resulting in an overall decreased ANF spontaneous activity (*Liberman, 1990*). Finally, when 1-methyl-4-phenyl-1,2,3,6-tetrahydropyridine (MPTP) was used to disrupt dopaminergic LOC innervation in guinea pig, spontaneous ANF activity was also reduced (*Le Prell et al., 2014*). This result could be explained by MPTP also targeting dopaminergic/cholinergic LOC neurons, and thereby reducing the cholinergic excitatory input onto ANFs.

## Activity-dependent plasticity in multi-transmitter neurons

The existence of multi-transmitter neurons has been increasingly recognized in the central nervous system, including the combinations of GABA/ACh, GABA/Glutamate, GABA/DA, DA/Glutamate (*Granger et al., 2017*; *Hnasko and Edwards, 2012*; *Tritsch et al., 2016*). The use of multiple neurotransmitters expands the range of synaptic coding that individual neurons can provide. LOC fibers in the auditory pathway now add DA/ACh to this increasing repertoire of transmitter combinations. Here, sensory stimulation modulates dopamine release from otherwise cholinergic efferent fibers. This process adds an additional layer of complexity to the behavior of these multi-transmitter LOC neurons, as the co-existence of the transmitter systems are stimulus and time-dependent.

Neurotransmitter switching has been reported in several brain areas of the adult nervous system, however, in most instances, it was artificially induced or occurred in a diseased state (*Spitzer, 2015*; *Spitzer, 2017*). The first example of adult neuron neurotransmitter switching in response to sensory stimuli has been reported in the paraventricular and periventricular nucleus of the hypothalamus, where neurons switch their neurotransmitter from somatostatin to dopamine (TH$^+$) after exposure to extended photoperiods (*Dulcis et al., 2013*; *Meng et al., 2018*). The current study provides another example of sensory induced changes in neurotransmitter identity. However, here instead of a complete pre- and postsynaptic switch from one neurotransmitter system to another, the LOC efferent terminals that gain TH expression seem to retain their cholinergic identity (*Figure 5—figure supplement 1A*) and postsynaptic dopamine receptors seem to be expressed constantly on most ANF endings (*Maison et al., 2012*). This arrangement allows the ANFs to respond immediately whenever DA is released. Therefore, rather than switching, LOC fibers operate with gain and loss of an additional neurotransmitter. Nevertheless, it remains to be investigated whether the expression of dopamine receptors will increase when dopamine release from the LOC fibers is chronically elevated. Such a mechanism has been observed in the midshipman fish inner ear, where seasonal changes of dopamine receptor expression on hair cells provide the main regulator for dopaminergic modulation of ANF activities (*Perelmuter et al., 2019*). Future studies are needed to further elucidate the complex feedback modulation of the first synapse in the auditory pathway.

# Materials and methods

**Key resources table**

| Reagent type (species) or resource | Designation | Source or reference | Identifiers | Additional information |
|---|---|---|---|---|
| Strain (*M. musculus*) | C57BL/6J WT | Jackson Laboratory | Stock #: 000664 RRID:IMSR_JAX:000664 | |
| Strain (*Rattus norvegicus domestica*) | Sprague Dawley rat | Charles River | RGD Cat# 734476 RRID:RGD_734476 | |
| Genetic reagent (*M. musculus*) | ChAT$^{iresCre}$ | Jackson Laboratory | Stock #: 028861 RRID:IMSR_JAX:028861 | |
| Genetic reagent (*M. musculus*) | Th$^{2A-CreER}$ | PMID: 28041852 | MGI: 98735 | Dr. David Ginty (Harvard Medical School) |

*Continued on next page*

*Continued*

| Reagent type (species) or resource | Designation | Source or reference | Identifiers | Additional information |
|---|---|---|---|---|
| Genetic reagent (*M. musculus*) | Ai3 (B6.Cg-*Gt(ROSA) 26Sor*<sup>tm3(CAG-EYFP)Hze</sup>/J) | Jackson Laboratory | Stock #: 007903 RRID:IMSR_ JAX:007903 | |
| Genetic reagent (*M. musculus*) | Ai9 (B6.Cg-*Gt(ROSA) 26Sor*<sup>tm9(CAG-tdTomato)Hze</sup>/J) | Jackson Laboratory | Stock #: 007909 RRID:IMSR_ JAX:007909 | |
| Antibody | Polyclonal rabbit anti-TH | Millipore | Cat# 657012–15 UG, RRID:AB_566341 | 1:500 – 1:1000 |
| Antibody | Polyclonal goat anti-GFP | Sicgen | Cat# AB0020-200; RRID:AB_2333099 | 1:5000 |
| Antibody | Polyclonal goat anti-ChAT | Millipore | Cat# AB144P RRID:AB_2079751 | 1:20 – 1:50 |
| Antibody | Polyclonal guinea pig anti-VAChT | Millipore | Cat# AB1588, RRID:AB_11214110 | 1:500 – 1:1000 |
| Antibody | Polyclonal goat anti-mCherry | Sicgen | Cat# AB0040-200; RRID:AB_2333092 | 1:1000 |
| Antibody | Monoclonal mouse anti-Myosin VIIa | DSHB | Cat# MYO7A 138–1, RRID: AB_2282417 | 1:200 - 1:500 |
| Chemical compound, drug | Dopamine Hydrochloride | Sigma-Aldrich | Cat# H8502 | |
| Chemical compound, drug | Sodium L-Ascorbate | Sigma-Aldrich | Cat# A7631 | |
| Chemical compound, drug | Tetrodotoxin | Tocris | Cat# 1078/1 | |
| Chemical compound, drug | Tetrodotoxin Citrate | Tocris | Cat# 1069/1 | |
| Chemical compound, drug | Tamoxifen | Sigma-Aldrich | Cat# T5648 | |
| Chemical compound, drug | Corn Oil | Sigma-Aldrich | Cat# C8267 | |
| Software, algorithm | ZEN Blue/Black | Zeiss | RRID:SCR_013672 | |
| Software, algorithm | ImageJ | NIH | RRID:SCR_003070 | |
| Software, algorithm | ImageJ plugin: Measure_line | Mass Eye and Ear | | https://research. meei.harvard.edu/ Otopathology/3dmodels/ other_tools.html |
| Software, algorithm | Fiji | http://fiji.sc | RRID:SCR_002285 | |
| Software, algorithm | MATLAB | MathWorks | RRID:SCR_001622 | |
| Software, algorithm | Adobe Photoshop CS6 | Adobe | RRID:SCR_014199 | |
| Software, algorithm | Adobe Illustrator CS6 | Adobe | RRID:SCR_014198 | |

*Continued on next page*

*Continued*

| Reagent type (species) or resource | Designation | Source or reference | Identifiers | Additional information |
|---|---|---|---|---|
| Software, algorithm | R Project for Statistical Computing | Open Source | RRID:SCR_001905 | |
| Software, algorithm | RStudio | Open Source | RRID:SCR_000432 | |
| Software, algorithm | SPSS Statistics 25 | IBM | RRID:SCR_002865 | |
| Software, algorithm | SigmaPlot | Systat Software Inc | RRID:SCR_003210 | |
| Software, algorithm | Origin 9.1 | OriginLab | RRID:SCR_014212 | |
| Software, algorithm | MiniAnalysis | Synaptosoft | RRID:SCR_014441 | |
| Software, algorithm | STATA | StataCorp | RRID:SCR_012763 | |
| Software, algorithm | pClamp 9.2 - Clampex | Molecular Devices | RRID:SCR_011323 | |
| Software, algorithm | pClamp 9.2 -Clampfit | Molecular Devices | RRID:SCR_011323 | |
| Software, algorithm | BatSound Touch Lite | Pettersson Elektronik | | |
| Software, algorithm | Adobe Audition CC 2018 | Adobe | RRID:SCR_015796 | |

## Mice and rats

All the experiments were performed on mice except for electrophysiological recordings that were performed on rats. Mice (WT strain C57BL/6J, Jackson Laboratory) and Rats (Sprague Dawley, Charles River Laboratories) of either sex were used in the experiments indiscriminately. For initial experiments (*Figures 2* and *3*), mice were raised in the institutional vivarium. For sound exposure experiments (*Figures 4* and *5*), mice were raised in a 'low-noise' satellite vivarium. Rats were raised in the institutional vivarium (*Figure 6*).

## Transgenic mouse models

Generation and genotyping of Cre driver lines *Chat*$^{iresCre}$ (*Rossi et al., 2011*), *Th*$^{2A-CreER}$ (*Abraira et al., 2017*) and Cre-dependent reporter mouse lines (Ai3 and Ai9, Allen Brain Institute) (*Madisen et al., 2010*) have been previously described. The tamoxifen injection procedure for the inducible Cre-loxP system when using the *Th*$^{2A-CreER}$ mouse line is described below. All transgenic mouse lines were either obtained on pure C57BL/6J background or bred in-house for at least nine generations with C57BL/6J WT mice before breeding for sound exposure experiments.

## Cochlear whole-mount immunofluorescence

Cochleas of one-week to three-month-old mice were harvested, perfused through the round and oval windows with 4% paraformaldehyde (Electron Microscopy Sciences), rinsed in phosphate buffered saline (PBS) and fixed for ~1 hr at room temperature (RT). Cochleas were carefully microdissected in PBS. Due to the presence of bones in cochleas from older mice, some partially dissected cochleas were decalcified in 0.2 M ethylenediaminetetracetic acid (EDTA) in PBS for 1–2 days before further processing. To achieve better penetration of primary antibodies, a fast freeze/thaw step in 30% sucrose was included occasionally. Whole-mount cochlear preparations were first incubated at RT in a blocking and permeabilizing buffer (PBS with 10% normal donkey serum, 0.5% Triton X-100) for 1–2 hr. Preparations were then incubated in primary antibody diluted in PBS containing 5% normal donkey serum, 0.25% Triton X-100% and 0.01% NaN$_3$ for ~42 hr at RT. Samples were rinsed

three times with PBS before incubation with the appropriate secondary antibody diluted 1:1000–2000 in PBS containing 5% normal donkey serum, 0.25% Triton X-100 at RT for 2 hr. Preparations were again rinsed three times in PBS before mounting on glass slides in FluorSave mounting medium (Calbiochem). All incubations and rinses were performed on a rocking table at RT. See Key Resources Table for a list of primary antibodies used in this study.

## Mapping of terminal region locations along the cochlear coil

The locations of 'terminal regions' with TH$^+$ bouton endings were mapped along the cochlear coil using the Measure_line ImageJ plugin (Massachusetts Eye and Ear Infirmary) by reconstructing the whole cochlear spiral from dissected pieces of cochlear coils. The logarithmic axis representing ANF characteristic frequency ($f$) was constructed using the formula d (%)=100 – (156.5 + 82.5 × log($f$)), based on *Müller et al. (2005)*. For cochleas shown in *Figure 2*, the terminal regions were identified by examining the reconstructed whole cochlea image made from tiles of confocal images taken with a 40x objective. For sound exposure experiments, terminal regions were identified by examining panorama confocal images of individual cochlear pieces taken with a 10x objective lens. In initial trials, the terminal regions were identified by using a higher magnification objective lens while viewing at the microscope and marking the location on the images taken with the 10x objective lens. Both methods gave comparable results.

## Brainstem sectioning and immunostaining

Mice were given an overdose of 50 mg/ml sodium pentobarbital and perfused transcardially with PBS followed by 4% paraformaldehyde. The brain tissue was then removed from the skull bones and post-fixed in 4% paraformaldehyde overnight at 4˚C. The caudal portion of the brain tissue was trimmed off to allow it to sit flat on its coronal plane and placed in a well coated with petroleum jelly. 5 ml of gel albumin mixed with 0.4 ml of 5% glutaraldehyde and 1 ml of 37% paraformaldehyde was placed into the well containing the brain tissue and allowed to harden for 30 s. The brain was mounted on the vibrating microtome (Vibratome 1000 Plus) with superglue and sectioned in the coronal plane into 50 μm slices. The brain slices were first incubated at RT in a blocking and permeabilizing buffer (PBS with 10% normal donkey serum, 0.5% Triton X-100) for 1–2 hr. The brain slices were then incubated with primary antibodies diluted in PBS containing 5% normal donkey serum, 0.25% Triton X-100 for two days in a cold-room at 4˚C on a shaker. After washing with PBS, the brain slices were incubated with secondary antibodies diluted in PBS containing 5% normal donkey serum, 0.25% Triton X-100 for 1–2 hr at RT on a shaker. The brain slices were then washed again with PBS, before mounting on glass slides in Fluoromount-G mounting medium (SouthernBiotech).

## Quantifying the number of TH$^+$ LSO intrinsic neurons

TH$^+$ LOC intrinsic neurons were quantified by visual inspection of confocal stacks of each brainstem slice. As a test, we circled all the TH$^+$ LOC intrinsic neurons found on two adjacent brainstem slices. Overlaying the images from two consecutive brainstem slices showed non-overlapping neurons, suggesting that it is unlikely to count one TH$^+$ LOC intrinsic neuron twice on consecutive brainstem slices. Therefore, a simple summation of the numbers found on each brainstem slice is appropriate for quantifying the total number of TH$^+$ LOC intrinsic neurons.

## Image acquisition

Fluorescence images were acquired using a LSM 700 confocal microscope (Zeiss) with a Fluar 10x/0.50 M27 objective, a LCI Plan-Neofluar 25x/0.8 Imm Korr DIC M27 objective and a Fluar 40x/1.30 Oil M27 objective using the ZEN black 2011 software. The pinhole was set at one airy unit. The size of optical sections was determined by stepping at half the distance of the theoretical z-axis resolution (the Nyquist sampling frequency). Images were acquired in a 1024 × 1024 raster. Images are presented as maximum intensity projections through a subset of the collected optical stacks. Some images were processed in ImageJ or Fiji without deconvolution, filtering, or gamma correction.

## Tamoxifen injections

Tamoxifen freebase (Sigma-Aldrich) was prepared in corn oil (Sigma-Aldrich) at 10 mg/ml and sonicated at RT for 0.5–1 hr until no precipitations were visible. This solution was stored at 4˚C for 5–7

days protected from light. Tamoxifen solution was administrated intraperitoneally (i.p.) or through gavage for a total amount of 0.2–2 mg for each injection. To achieve sparse labeling of TH-expressing neurons, a single dose of 0.2–0.5 mg tamoxifen was administrated.

## Sound exposure

Octave-band noise was generated with digital-to-analog converters and gated with electronic switches (Tucker-Davis Technologies). Stimulus levels were controlled by programmable attenuators (Tucker-Davis Technologies) and an audio amplifier (Crown Audio). Stimulus waveforms were transduced by two overhead high-frequency speakers (Pyramid). The overhead location of the speaker minimized the effects of head orientation on sound energy propagating to the tympanic membrane. Speakers were calibrated with the same microphone and software used for the auditory brainstem response measurements (see below). Additional checks on the sound levels were performed using a sound level meter fitted with a ½" free field microphone (Larson-Davis) just prior to each sound exposure.

During the presentation of noise, mice were placed inside a mesh cage with food pellets and water gel *ad libitum*. The cage was situated on top of a rotating platform making counter-clock or clockwise rotations 30 s/step to make sure even exposure to the sound field. Each step was 1.8 degrees and happened abruptly. Sound levels at test frequencies varied by no more than 10 dB within the listening area. The whole platform was isolated from extraneous environmental sounds by a sound-attenuation chamber. The inner walls of the chamber were lined with anechoic foam (Sonex). For most of the trials, control littermates were placed inside a mesh cage on top of a simultaneously rotating platform with the same built outside of the sound-attenuation chamber and exposed to ambient noise levels in the procedure room during the sound exposure session. For two trials, control littermates were either placed in the shared housing cage outside the sound-attenuation chamber or placed on the rotating platform inside the sound-attenuation chamber during daytime without noise exposure.

Mice were exposed to a 1-octave band noise centered at 12 kHz. The 2 hr 110 dB SPL exposure began with an initial moderate level for 15 s, followed by an interim level for 15 s to finally reach the maximum level of ~110 dB SPL. The maximum sound exposure level was maintained for 2 hr. The exposure was performed during the daylight cycle at the vivarium. A less damaging sound exposure used a 1-octave band noise centered at 12 kHz at ~90 dB SPL. The exposure lasted 12 hr each session. All of the mice were exposed five consecutive nights (approximately 9 PM – 9 AM), except for one that was exposed for three consecutive nights, though the whole dataset was collectively referred to as 5 × 12 hr 90 dB SPL exposure.

For sound exposure and control experiments, both wild-type and transgenic mice were bred on a pure C57BL/6J background and were raised in a low-noise satellite vivarium. The sound exposure started at 7–8 weeks of age. Mice were sacrificed 7–10 days after the beginning of the sound exposure. Cochlear and brainstem samples were harvested for immunostaining experiments. Mice used in the sound exposure or control experiments were not used in any previous procedures or sound exposures.

## Vivarium sound level measurements

Sound levels in the 'low noise' vivarium and the large-capacity institutional vivarium were recorded using a data-logging sound level meter with 1/3 octave band measurement capabilities and ½" free field microphone (Larson-Davis). Ultrasound range was measured in each room using a M500-384 USB Ultrasound Microphone (Pettersson Elektronik) linked to a laptop running the BatSound Touch Lite software. 5 min recordings were made simultaneously with the sound level meter and the ultrasound microphone. Such recordings were made on 4–5 days at slightly different locations inside each vivarium, where the mice were housed. All the recordings were made during daytime. Frequency analysis of the recorded sound was performed using Adobe Audition CC2018 (Adobe).

## Auditory brainstem response measurements

Recording procedures were similar to those previously described (*Lauer and May, 2011*; *Lina and Lauer, 2013*; *McGuire et al., 2015*). Mice were anesthetized with 100 mg/kg ketamine and 20 mg/kg xylazine through intraperitoneal injection and placed on an electronically controlled heating pad

inside a small sound-attenuating chamber 30 cm away from a Fostex speaker in front of the animal. Auditory brainstem responses (ABRs) were differentially recorded from the scalp using subcutaneous platinum needle electrodes (G.R.A.S.) placed over the left bulla and at the vertex of the skull, with a ground electrode inserted into the leg muscle. Responses were amplified 300,000 times (ISO-80, World Precision Instruments) and bandpass filtered from 300 to 3,000 Hz (Krohn-Hite). Auditory stimuli were generated at a 200 kHz sampling rate, attenuated to control presentation levels (Tucker-Davis Technologies, PA5), and amplified (Parasound) before being passed to a calibrated free-field speaker (Fostex). Stimulus protocols were implemented on programmable real-time processors (Tucker-Davis Technologies, RX6) using a custom MATLAB program (*Ngan and May, 2001*). Responses were averaged over 300 stimulus presentations. Stimuli were clicks and 5 ms tone pips at 8, 12 16, 24, 32 kHz with a rise/fall time of 0.5 ms played at a rate of 20 or 30 repetitions/s. Stimuli were calibrated with a ¼″ Bruel and Kjaer microphone placed at the location normally occupied by the mouse's head during testing using a custom MATLAB (Mathworks) based program.

Thresholds and suprathreshold responses were measured by presenting a descending series of stimulus levels beginning with −10 dB of the maximum possible output of the speaker and continuing in 5- or 10 dB steps until no response could be discerned from the noise. Threshold was defined as the sound level at which the peak-to-peak ABR amplitude (any wave) was two standard deviations above the average level of a 5 ms window of baseline noise collected at the end of a 30 ms recording epoch. Responses to clicks were measured first to ensure an optimal signal-to-noise ratio and the presence of at least four distinct peaks. Responses to tones were measured in a pseudorandom order.

## Electrophysiology

Recordings were performed on acutely excised cochlear preparations from Sprague Dawley rats (Charles River Laboratories) at room temperature (22–25℃). Recording pipettes were fabricated from 1 mm borosilicate glass (WPI). Pipettes were pulled with a multistep horizontal puller (Sutter), coated with Sylgard (Dow Corning) and fire polished. Drug application was mediated by whole-bath perfusion or a gravity-driven flow pipette (100-µm-diameter opening) placed near the row of IHCs and connected with a VC-6 channel valve controller (Warner Instrument).

## ANF recordings

Postnatal day (P) 15–31 SD rats were used for ANF recordings. Loose-patch extracellular and whole-cell patch clamp recordings in current and voltage clamp on the dendritic endings of ANFs were performed as described before (*Grant et al., 2010*; *Wu et al., 2016*). Pipette resistances were 9–15 MΩ for ANF recordings. ANF recordings were acquired using pCLAMP 9.2 or pCLAMP 10.2 software (Molecular Devices) in conjunction with a Multiclamp 700A or Multiclamp 700B amplifier (Molecular Devices). The signal was low pass filtered at 10 kHz and digitized at 50 kHz with a Digidata 1322A (Molecular Devices).

Extracellular solutions for ANF recordings contained (in mM): 5.8 KCl, 144 NaCl, 0.9 $MgCl_2$, 1.3 $CaCl_2$, 0.7 $NaH_2PO_4$, 5.6 glucose, 10 HEPES, pH 7.4 (NaOH), 300 mOsm. In a subset of ANF voltage-clamp recordings, $K^+$ concentration is elevated to 15 mM in substitution of $Na^+$ to increase presynaptic release. Pipette solution for extracellular loose-patch recordings contained the extracellular solution. Pipette solutions for whole-cell ANF recordings contained (in mM): 20 KCl; 110 K-methanesulfonate; 5 $MgCl_2$; 0.1 $CaCl_2$; 5 EGTA; 5 HEPES; 5 $Na_2ATP$; 0.3 $Na_2GTP$; 5 $Na_2$ phoshocreatine; pH 7.2 (KOH), 290 mOsm or 135 KCl, 3.5 $MgCl_2$, 0.1 $CaCl_2$, 5 EGTA, 5 HEPES, 0–4 $Na_2ATP$, 0–0.2 $Na_2GTP$, pH 7.2 (KOH), 290 mOsm.

For ANF current clamp recordings, bridge balance and pipette capacitance neutralization were performed. For ANF voltage clamp recordings, holding potentials were between −99 and −84 mV. All recordings that had a leak current <350 pA, except for one recording that had a leak current up to 450 pA. Both current and voltage clamp recordings were corrected posterior for measured liquid junction potentials: 4 mV for potassium-based solution and by 9 mV for methanesulfonate-based solution. Series resistance Rs was not compensated for in voltage clamp recordings.

## Drugs

DA solutions were prepared daily from dopamine hydrochloride powder (Sigma) and protected from light during the experiments. 1 mM DA was used in ANF loose-patch extracellular and current-clamp recordings, 1–2 mM DA was used in ANF voltage-clamp recordings. 0.01–0.1% (w/v) sodium ascorbate was supplemented in the DA solution as an antioxidative agent, except for two ANF recordings. For control experiments, DA but not sodium ascorbate was omitted in the Control solution. In a subset of voltage-clamp recordings, 1–2 µM TTX (Tocris) was present to block APs.

## Timeline of drug application and analysis of firing and release rates

30–60 s are typically needed for a drug to wash into the tissue, and this was the minimal wait time before analyzing dopamine effects. For cochlear afferent recordings, the rate of EPSC/EPSP and AP fluctuates in a power-law fashion throughout the recording, even in control condition, as described in *Wu et al. (2016)*. Therefore, it is essential, to record for long enough in control, in DA, and after washout, to be able to separate random fluctuations from drug effects. This is why at least 30 s of recording were analyzed before, in and after DA.

To study the effects of DA, segments of recordings were selected for analysis. For extracellular loose-patch recordings of ANFs, 2 min before DA application, 2 min before the end of DA application, and 2 min after 2 min of washout, were taken as the 'before', 'DA' and 'after' time windows, respectively. The control experiments were performed and analyzed in the same way. The total application time for DA or Control solution was 3–5 min.

For whole-cell recordings of ANFs, 30 s to 2 min of recording before DA application was selected for 'before', 1 or 2 min of recording after at least 30 s of DA application was selected for 'DA', 30 s to 2 min of recording after at least 1.5 min of wash-out was selected for 'after'. The total application time of DA varied between 1.5–6 min.

## Synaptic event detection and graphing

Events in extracellular loose-patch and whole-cell current clamp recordings were detected using a routine in MiniAnalysis and subsequently accepted by eye. In current clamp, each peak detected was identified as an event for quantification of synaptic frequency. AP and EPSP were identified by their amplitude distribution. AP threshold was detected by several methods. The first method found the point of maximum slope of the first derivative of the rising phase of the AP using a customized MATLAB routine (*Wu and Chan, 2019*). For most of the APs in the ANF dendrite recordings, this method could successfully identify the AP threshold. In case this method failed to locate the AP threshold, a second method that found the maximum curvature point on the rising phase of the AP was used. If the second method still could not identify the AP threshold, the AP analysis routine in the MiniAnalysis (Synaptosoft) was used.

Events in ANF voltage clamp recordings were detected by threshold search event detection method of Clampfit (Molecular Devices), following a manual adjustment of baseline current. The end of an event was defined as the current that came back to the pre-set value from the baseline. Therefore, an event could contain several closely spaced peaks.

Data were plotted using Excel 2016 (Microsoft), Origin 9.1 (OriginLab), RStudio, MATLAB (MathWorks), and Illustrator (Adobe). For representative traces, the data points were reduced by decimation to a 5 or 10 kHz sampling rate.

## Statistics

Statistical analyses were performed with SPSS Statistics 25 (IBM), STATA (StataCorp) or R studio (R Core Team). Graphical representation of the quantification is defined in Figure Legend, with the definition of n and information about the statistical tests, unless otherwise states in the Result. Error bars can be either standard deviation (SD) or standard error of the mean (SEM), as specified in Figure Legend. Statistical significance is defined as: n.s. (not significant) $p>0.05$, $*p<0.05$, $**p<0.01$, $***p<0.001$. For the quantification of the $TH^+$ terminals and $TH^+$ LOC neurons, the researchers that performed the majority of the analysis were blinded to the experimental condition during data quantification.

## Acknowledgements

This work is supported by a National Institute on Deafness and Other Communication Disorders grant P30 DC005211 to the Center for Hearing and Balance at Johns Hopkins. Additional support for the statistical analysis is from the National Center for Research Resources and the National Center for Advancing Translational Sciences (NCATS) of the National Institutes of Health through Grant Number 1UL1TR001079. We thank Brian McGuire for contributing to sample processing and data analysis. We thank Kristina Jones for contributing to the quantification of the TH$^+$ LOC intrinsic neurons. We thank Timothy Phelps for making the illustrations in *Figure 1*.

## Additional information

### Funding

| Funder | Grant reference number | Author |
| --- | --- | --- |
| National Institute on Deafness and Other Communication Disorders | R01DC006476 | Elisabeth Glowatzki |
| National Institute on Deafness and Other Communication Disorders | R01DC012957 | Elisabeth Glowatzki |
| National Institute on Deafness and Other Communication Disorders | R01DC017620 | Amanda M Lauer |
| National Institute on Deafness and Other Communication Disorders | R01DC016641 | Amanda M Lauer |
| David M. Rubenstein Fund for Hearing Research | | Amanda M Lauer |
| Capita Foundation | | Amanda M Lauer |
| Korea Health Industry Development Institute | Korea Health Technology R&D Project HI17C0952 | Eunyoung Yi |

The funders had no role in study design, data collection and interpretation, or the decision to submit the work for publication.

### Author contributions

Jingjing Sherry Wu, Conceptualization, Data curation, Software, Formal analysis, Investigation, Methodology, Writing - original draft, Writing - review and editing; Eunyoung Yi, Conceptualization, Data curation, Funding acquisition, Investigation, Methodology, Writing - review and editing; Marco Manca, Formal analysis, Investigation, Writing - review and editing; Hamad Javaid, Formal analysis, Investigation, Methodology; Amanda M Lauer, Resources, Supervision, Funding acquisition, Methodology, Writing - review and editing; Elisabeth Glowatzki, Conceptualization, Resources, Supervision, Funding acquisition, Methodology, Project administration, Writing - review and editing

### Author ORCIDs

Jingjing Sherry Wu (iD) https://orcid.org/0000-0001-9625-1661
Amanda M Lauer (iD) http://orcid.org/0000-0003-4184-7374
Elisabeth Glowatzki (iD) https://orcid.org/0000-0003-3135-658X

### Ethics

Animal experimentation: This study was performed in strict accordance with the recommendations in the Guide for the Care and Use of Laboratory Animals of the National Institutes of Health. All procedures concerning animals were performed in accordance with animal protocols approved by the Johns Hopkins University Animal Care and Use Committee (IACUC). The Johns Hopkins University Office of Laboratory Animal Welfare (OLAW) Assurance number is A-3272-01.

Decision letter and Author response
Decision letter https://doi.org/10.7554/eLife.52419.sa1
Author response https://doi.org/10.7554/eLife.52419.sa2

## Additional files

### Supplementary files
• Transparent reporting form

### Data availability
All data generated or analyzed during this study are included in the manuscript and supporting files. Source data file has been provided for Figure 4B.

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
