## [Decision Letter]

**Acceptance summary:**

Efferents project to the inner ear, and serve a crucial role in gain control and filtering of behaviorally relevant sound signals. There are two types of olivocochlear efferents, with most previous research focused on the medial olivocochlear neurons that synapse onto outer hair cells. Much less is known about the function of lateral olivocochlear neurons. Their synapses are strategically located on auditory nerve endings, directly where they contact hair cells, allowing them to affect firing rates and neural coding at the first synapse in the auditory pathway. Indirect evidence suggests that lateral olivocochlear neurons respond to sound and form a three-neuron feedback loop, the 'LOC acoustic reflex'. This paper shows that sound exposure induces dopamine synthesis in the LOC neurons and suggests dopaminergic modulation of auditory nerve fibers is both activity-dependent and dynamic. The authors also provide support for the hypothesis that cholinergic and dopaminergic transmitter systems act together.

**Decision letter after peer review:**

Thank you for submitting your article "Sound exposure dynamically induces dopamine synthesis in cholinergic LOC efferents for feedback to auditory nerve fibers" for consideration by *eLife*. Your article has been reviewed by three peer reviewers, and the evaluation has been overseen by Andrew King as the Senior Editor. The following individuals involved in review of your submission have agreed to reveal their identity: Michael Evans (Reviewer #1); Charles Liberman (Reviewer #2); J. Christopher Holt (Reviewer #3).

The reviewers have discussed the reviews with one another and the Reviewing Editor has drafted this decision to help you prepare a revised submission.

Summary:

This study provides new insights into the plasticity of the lateral olivocochlear efferent system, using a combination of reporter mice, immunohistochemistry and in vitro electrophysiology. The authors use tyrosine hydroxylase levels as a proxy for dopamine synthesis in olivocochlear neurons to provide strong support for the presence of dopamine within some cholinergic lateral cochlear efferent nerve fibers. These two neurotransmitters had been previously shown to have opposing effects on auditory nerve fiber responses, and their co-existence in the same nerve fibers could provide an up or down regulation of auditory nerve responses. The authors further show that this transmitter expression appears to be modulated by sound exposure history. Thus, the same neurons can use either acetylcholine or dopamine to either excite or inhibit the responses of their target neurons. The authors then supplement the morphological data with electrophysiological recordings in the rat cochlea from the bouton endings of auditory afferents on inner hair cells before and after the exogenous application of dopamine. In this preparation, they show that dopamine, when applied exogenously, likely acts on both presynaptic and postsynaptic targets to reduce both quantal rate and size.

Essential revisions:

Reviewers were concerned about the plotting and quantification of TH^+^ terminals along the cochlear spiral. The frequency axes in Figures 2 and 4 are linear with frequency, but cochlear frequency mapping transforms linear distance into log frequency along the mouse spiral. Thus, plots like Figure 4A and 4B are distorted, since the 19 kHz region is at the middle of the cochlear spiral. Please also consider that the frequency axis may be wrong, if the observers noted terminal locations in linear distance and then plotted them along a linear frequency axis with appropriate lowest and highest frequency values. The plot in Figure 4C may be inaccurate in its estimation of the "% of the cochlea with TH^+^ terminals". We suggest that you address these points, and add details to your Materials and methods.

---

## [Author Response]

Essential revisionsReviewers were concerned about the plotting and quantification of TH^+^ terminals along the cochlear spiral. The frequency axes in Figures 2 and 4 are linear with frequency, but cochlear frequency mapping transforms linear distance into log frequency along the mouse spiral. Thus, plots like Figure 4A and 4B are distorted, since the 19 kHz region is at the middle of the cochlear spiral. Please also consider that the frequency axis may be wrong, if the observers noted terminal locations in linear distance and then plotted them along a linear frequency axis with appropriate lowest and highest frequency values. The plot in Figure 4C may be inaccurate in its estimation of the "% of the cochlea with TH^+^ terminals". We suggest that you address these points, and add details to your Materials and methods.

We agree with the reviewers that in the original version of Figures 2 and 4, the frequency axis was inaccurately plotted on a linear scale.

We have updated these plots in Figure 2D, Figure 4A and 4B and figure legends. The locations of patches with TH^+^ terminals along the cochlear coil were determined using the measure_line imageJ plug-in developed at the Eaton-Peabody Laboratories (Mass. Eye and Ear Infirmary, Boston, MA). The axis of the plots now represents the linear distance along the cochlear spiral (in% ). Based on these measurements, Figure 4C accurately represents the% of cochlea covered with TH^+^ terminals. Secondly, in Figures 2D and Figure 4A, 4B, below the axis representing% cochlear spiral, a second axis is plotted for reference to relate locations along the cochlear spiral to the frequency coded.

In Results now read: “In these line plots, upper lines represent terminal regions and lower lines spiral regions. The apical cochlear tip was set at 0%, and the basal tip at 100% of cochlear length. Below the linear axis representing cochlear length, as reference a logarithmic map of ANF characteristic frequency is plotted, based on (Müller et al., 2005).”

In Materials and methods now read: “The locations of ‘terminal regions’ with TH^+^ bouton endings were mapped along the cochlear coil (0 to 100%, apex to base) using the Measure_line ImageJ plugin (Massachusetts Eye and Ear Infirmary) by reconstructing the whole cochlear spiral from dissected pieces of cochlear coils. The logarithmic axis representing ANF characteristic frequency (*f*) was constructed using the formula d (%) = 100 – (156.5 + 82.5 × log(*f*)), based on (Müller et al., 2005).”